# The Diversified O-Superfamily in *Californiconus californicus* Presents a Conotoxin with Antimycobacterial Activity

**DOI:** 10.3390/toxins11020128

**Published:** 2019-02-20

**Authors:** Johanna Bernáldez-Sarabia, Andrea Figueroa-Montiel, Salvador Dueñas, Karla Cervantes-Luévano, Jesús A. Beltrán, Ernesto Ortiz, Samanta Jiménez, Lourival D. Possani, Jorge F. Paniagua-Solís, Jorge Gonzalez-Canudas, Alexei Licea-Navarro

**Affiliations:** 1Departamento de Innovación Biomédica, CICESE, Carretera Ensenada-Tijuana 3918, Ensenada, BC C.P. 22860, Mexico; jbernald@cicese.edu.mx (J.B.-S.); figuerm@cicese.edu.mx (A.F.-M.); sduenas@cicese.edu.mx (S.D.); kecervanlu@gmail.com (K.C.-L.); mjimenez@cicese.edu.mx (S.J.); 2Departamento de Ciencias Computacionales, CICESE, Carretera Ensenada-Tijuana 3918, Ensenada, BC C.P. 22860, Mexico; abeltran@cicese.edu.mx; 3Departamento de Medicina Molecular y Bioprocesos, IBT, UNAM, Cuernavaca, Morelos, C.P. 62210, Mexico; erne@ibt.unam.mx (E.O.), possani@ibt.unam.mx (L.D.P.); 4Teraclon IDF, S.L., Parque Tecnológico de Madrid, Tres Cantos, Madrid, C.P. 28760, Espana; jpaniagua@teraclon.com; 5Laboratorio Silanes, S.A. de C.V., Ciudad de México, C.P. 11000, Mexico; jogonzalez@silanes.com.mx

**Keywords:** *Conus californicus*, *Californiconus californicus*, O-superfamily conotoxin, transcriptome, tuberculosis, anti-TB, antimycobacterial, multidrug-resistant tuberculosis

## Abstract

*Californiconus californicus*, previously named *Conus californicus*, has always been considered a unique species within cone snails, because of its molecular, toxicological and morphological singularities; including the wide range of its diet, since it is capable of preying indifferently on fish, snails, octopus, shrimps, and worms. We report here a new cysteine pattern conotoxin assigned to the O1-superfamily capable of inhibiting the growth of *Mycobacterium tuberculosis* (Mtb). The conotoxin was tested on a pathogen reference strain (H37Rv) and multidrug-resistant strains, having an inhibition effect on growth with a minimal inhibitory concentration (MIC) range of 3.52–0.22 μM, similar concentrations to drugs used in clinics. The peptide was purified from the venom using reverse phase high-performance liquid chromatography (RP-HPLC), a partial sequence was constructed by Edman degradation, completed by RACE and confirmed with venom gland transcriptome. The 32-mer peptide containing eight cysteine residues was named O1_cal29b, according to the current nomenclature for this type of molecule. Moreover, transcriptomic analysis of O-superfamily toxins present in the venom gland of the snail allowed us to assign several signal peptides to O2 and O3 superfamilies not described before in *C. californicus*, with new conotoxins frameworks.

## 1. Introduction

As a single member of its genera, *Californiconus californicus*, has shown how valuable it can be. *C. californicus*, a unique species inhabiting in coasts of California and Baja California, has filled niches often divided among multiple cone snail species. Particular feeding behaviors related them to organized/cooperative prey attacks [1], taking distance from conventional characteristics present in cone snails, including the eastern Pacific members. Indeed, molecular singularities put them in a special classification. As more peptides are characterized, it is more difficult to fit conotoxins described from *C. californicus* into the conventional established toxin classification, reflecting their evolutionary distance from the rest of the species. 

Approximately 26 different superfamilies have been characterized from cone snails, where the majority of peptide toxins expressed are mostly from A, M, and O-superfamilies. For *C. californicus*, no venom peptides have been identified belonging to the A and M-superfamilies. Instead, a strong influence of O-superfamily conopetides dominates in terms of their expression level and a number of isoforms, suggesting an important role in prey capture and/or defense [1,2,3]. In general, the O-superfamily has been subdivided in O1, O2, and O3; is composed of three cysteine frameworks VI/VII, XV, and XII; and is classified into δ, μO, ω, κ and γ with an extensive target repertoire in ion channels [3]. Until today, reports have only assigned the O1 gene superfamily to the venom repertoire of *C. californicus* [4].

The well-known potential of conotoxins for specific actions in their therapeutic targets has increased the interest in exploring other biomedical fields. Conotoxin research in *C. californicus* has exhibited various promising peptides that could be scaled up in biomedical research, like a transcription of pro-inflammatory cytokines, anti-parasitic effect and apoptosis activation in lung cancer [5,6,7]. In general, anti-bacterial activity in conotoxins has not been fully related to their potential, until recently with the first report associating conotoxins capable of inhibiting the growth of a reference strain of Mtb [8]. Notably, *M. tuberculosis* exhibits a complex membrane that has allowed it to evolve and remain as a major health problem affecting a third of the population worldwide. According to the World Health Organization (WHO), in 2017 there were 10 million new cases and 1.3 million deaths from tuberculosis (TB) around the world, highlighting strong implications of multi-drug resistant strains (MDR), specially Rifampicin resistance (RR-TB). For this reason, the WHO in the *Global Tuberculosis Report* has alerted as a priority to intensify research and innovation in new drug treatment to cut the risk of TB disease in the 1.7 billion people already latently infected [9].

In this report, we described a conotoxin named O1_cal29b, as new Cys framework member of the O1-superfamily, capable of inhibiting a reference strain of *M. tuberculosis* H37Rv, one MDR and one more MDR/RR-TB strain. In addition, we present the conotoxins belonging to the O2 and O3 superfamilies that were elucidated from the venom gland transcriptome of *C. californicus*.

## 2. Results

### 2.1. Purification of O1_cal29b

The reverse phase high-performance liquid chromatography (RP-HPLC) profile of the crude venom extract from *C. californicus* is shown in Figure 1. The strategic method used for purification was the isolation of the fraction with antimycobacterial activity (red arrow), performing subsequent purifications until obtaining a pure peptide (blue arrow). In the first evaluation, an antimycobacterial assay revealed activity only in 1 of 12 tested fractions (minutes 35 to 40, Appendix A). The active toxin was further purified using a slower gradient; the resulting peak of interest is indicated by a blue arrow in Figure 1.

### 2.2. Identification of cDNA Clone Encoding O1_cal29b Precursor

The active peptide was subjected to conventional N-terminal sequencing by the Edman degradation method (Atheris Laboratories). Partial information of the amino acids sequence was obtained, identifying 16 of 32 amino acids of the conotoxin sequence. Based on this information, a RACE strategy was employed to isolate the encoding gene. Degenerate primers were used to obtain the remainder of the amino acid sequence for the N-terminal leader, the propeptide region, the toxin-coding region, and the 3´UTR (Figure 2a). 

A 361-base cDNA was obtained through a combination of 3´- and 5´-RACE (Figure 2a). The structure of this cDNA has three segments: a coding region for a 22-residue signal peptide predicted by using the SignalP 4.1 server [10], a 21-residue propeptide, and a 32-residue mature peptide with an additional Gly residue which is a prerequisite for amidation of the C-terminal Ser. The cloned signal peptide of our conotoxin was found to be similar to the O1-superfamily, aligned in Figure 2b. The mature peptide sequence was highly similar to cl tx-4 referred to by Biggs [1], with a punctual difference in position 20 and 21; where their conotoxin has the amino acids -DK- while in ours, an inverted -KD- pair is shown, confirmed by Edman degradation analysis as well. The conotoxin sequence exhibit eight cysteines with the framework CCC-C-CC-C-C. This framework shown in reference [1] was not classified. Until now, a total of 28 cysteine scaffolds have been described [11]. Therefore, we propose for this framework the subsequent number 29, and thus our conotoxin has been named O1_cal29b with consecutive letter b due to we named cal29a, the conotoxin (cl tx-4) described by Biggs [1].

Otherwise, cDNA includes 3´UTR with 130 bases in which there is only one polyA addition signal (PAS), and the polyA tail contains only 10 adenines. It seems that this conotoxin does not support the conventional idea that with a greater number of PAS, a shorter polyA tail is expected. There is no 5´UTR sequence available. 

### 2.3. Transcriptomic Analysis for Members of O-Superfamily 

For the O1-superfamily, 44 complete sequences were identified from the *C. californicus* venom duct transcriptome (Table 1). Seventeen of them have been previously reported (Table 1a), and 25 more are just described in this work (Table 1b). O1_cal29b conotoxin, enlisted in second group, allowed us to confirm the position 20 and 21 for amino acids -KD-. The cysteine frameworks observed were the conventional VI/VII, I, XII, and we add framework 29 for our conotoxin.

Additionally, through transcriptomic analysis one conotoxin with a signal peptide related to the O2-superfamily (Table 1b) was found. This sequence has a novel framework C-C-CCC-C-C-CC not described before. For this reason, the novel conotoxin was assigned the consecutive framework number 30. Furthermore, 12 complete sequences related to the O3-superfamily were identified after the bioinformatic analysis. Eight conotoxins with XIV, VI/VII, and XXVII frameworks were found. Likewise, four mature peptides had the known scaffold arrangement of one Cys pair.

### 2.4. Antimycobacterial Susceptibility Assay

O1_cal29b was tested in an antimycobacterial susceptibility assay on the H37Rv strain (Figure 3a), showing a tendency to inhibit the strain growth at 3.5 μM. This result leads us to perform a minimal inhibitory concentration (MIC) test on two MDR strains. The peptide O1_cal29b was evaluated at a concentration range of 3.52–0.22 μM (Figure 3b,c). In the MDR-1 assay, a strain resistant to Streptomycin and Isoniazid was used; here, our conotoxin showed to be capable to inhibit growth at 0.22 μM as we can see in Figure 3b. For the MDR-2 assay, a strain resistant to three first-line drugs Isoniazid, Pyrazinamide and Rifampicin were used; in this case, O1_cal29b showed a 1.76 μM efficiency.

## 3. Discussion

*M. tuberculosis* is one of the most successful pathogens in human history and remains a global health challenge. Since the WHO declared tuberculosis to be a global health emergency in 1993, research efforts have focused on all aspects of the illness, highlighting the priority of intensified research and innovation in new drug treatment. The challenge includes ending the epidemic by 2030. However, tuberculosis remains a major health problem due to a combination of the current lack of effective therapy, prolonged treatment periods and extensive side effects of toxic chemotherapeutic agents, whereas drug-sensitive TB strains needs to be treated for six months of chemotherapy, using the current frontline drugs regimen. Treatment for Rifampicin-resistant TB (RR-TB) and MDR-TB is time consuming and requires more expensive and toxic drugs to cure the illness. In this case, treatment includes at least 18 to 24 months of therapy with four to six drugs [12], with a treatment success rate around 55%, globally.

Except for the novel anti-TB drug Bedaquiniline approved by the United States Food and Drug Administration (FDA) in 2012, no novel drugs have been marketed for TB in the past 60 years. Ideally, anti-TB compounds must offer novel modes of action and special structures to avoid mycobacterial resistance. New alternatives in this regard are the development, design and improvement of antimicrobial peptides (AMPs) derived from bacteria, fungi, plants and animals [13]. Some mechanisms of action proposed are related to membrane receptors or channels involved in the nutrient intake of *M. tuberculosis*, that serve as an essential process for survival and colonization of the human respiratory tract [14,15].

While most use of conotoxins has been focused on pain management [16]; other cysteine-rich proteins/peptides isolated from the snake, scorpion, and spider venoms have shown strong antimicrobial potency against bacterial, fungal and yeast infections and are a promising source for therapeutic agents [17]. Before the discovery of I1_xm11a, a native peptide with anti-TB activity [8], an effective antimicrobial activity from conopeptides was shown for Lo6/7a conotoxin from *C. longurion* [18], as well as with the engineering of ω-conotoxin MVIIA into cyclic analogs [19]. 

*C. californicus* with divergent conotoxins present in its venom is an excellent candidate to explore the possibility of finding new molecules capable of inhibiting tuberculosis bacilli, with special attention on MDR-TB strains. In our report, we found a conotoxin named O1_cal29b with activity against H37Rv reference strain at 3.5 μM. Our results indicate that Mtb was slightly susceptible at the same concentrations order of two first-line drugs tested (INH 0.90 μM and EMB 4.9 μM). In this case, we did not assay MIC for the H37Rv strain because of the limited material; O1_cal29b could be effective at lower concentrations. However, when we tested O1_cal29b on MDR-1, a strain resistant to INH and SM, we detected a 0.22 μM efficiency inhibiting bacterial growth. Moreover, it was also effective on MDR-2, a strain resistant to three first-line drugs such as INH, RIF and PZA, where it demonstrated a MIC of 1.76 μM, making it a competitive candidate for further in vitro assays.

Based on the sequence of O1_cal29b, it is difficult to assign a possible pharmacological target, because there no other member with the same framework has been previously described. As all conotoxins reported, our conotoxin could be modulating a receptor or channel, in this case on bacterial membrane. As we know, the O1 superfamily is one of the largest groups composed of six cysteine frameworks (VI/VII, XII, I, XIV, IX, XVI), with extensive target repertoire over ion channel [20]. In this report, we included to O1_cal29b in a new framework not described before in this superfamily.

It is complicated to elucidate the reason why we did not find the cl tx-4 sequence reported previously by Biggs, since in the three experimental analyses (Edman degradation, RACE, and venom transcriptome) only amino acids -KD- in positions 20 and 21 were confirmed. However, we do not discard the possibility that a related family of conotoxins exists, which O1_cal29b and cl tx-4 (or cal29a) are part of. 

Until today, only members of O1-superfamily had been assigned to *C. californicus* venoms. However, this work describes the presence of O2 and O3 superfamilies in its venom repertoire. In the transcriptomic analysis, we have found one conotoxin that shares sequence homology in their signal peptide with the known gene superfamily O2 precursor. Nonetheless, it is important to consider that the propeptide and predicted mature peptide regions were different, and that the mature peptide of 58 aminoacids displays five disulfide bonds, not associated to this superfamily before. Conventionally, the O2 superfamily can be divided into two main groups, those with cysteine framework VI/VII and the single disulfide-containing contryphans [21]. Here, we propose to include one new Cys framework member with an arrangement not described before (C-C-CCC-C-C-C-CC). For this framework, we proposed to assign number 30. We are aware that we should proceed with caution when classifying this conotoxin in this superfamily, but everything so far indicates that it belongs to it. Indeed, this is the first time where the O3-superfamily was found to be present in venom ducts on *C. californicus*. Twelve complete sequences were related to this superfamily with frameworks previously assigned like VI/VII, XIV and XXVII; the last one was only reported by Hu et al [22]. Interestingly, we have found four toxins with the framework C-C, arrangement only described for contryphan toxins (members of O2 superfamily), but in this case, with a signal peptide assigned to the O3 superfamily.

It is possible that the reason why these superfamilies were not found in previous analyses is that a transcriptomic approach has more coverage of the conotoxins present in the venom than a cDNA library. Moreover, conotoxin expressions vary depending on the circumstances since the animal changes the composition of its venom depending whether they are using it for preying, defending themselves, or eating; this last one being so diverse with the generalist diet of *C. californicus*. Furthermore, studies suggest differential expression profiles in different segments of venom ducts, where an analysis on *C. geographus* indicates that proximal, proximal central and distal central mostly express O1 superfamily, but distal segment contains a much more diversified spectrum of conotoxins, including a significant proportion of O2 and O3 superfamilies [22]. Therefore, this could be an important factor to explain why O2 and O3 superfamilies have not been reported previously in *C. californicus*. However, it is important to highlight the presence of O2 and O3 superfamilies in the transcriptome of this species but that the role of these conotoxins as prey/predator members of this marine snail still remain unknown to us.

## 4. Conclusions

O1_cal29b is a competitive candidate for further *in vitro* assays to lead its potential as an antifimic molecule or to explore novel pharmacological targets capable of inhibiting multidrug-resistant *Mycobacterium tuberculosis*. 

O1_cal29b presents the same sequence as cl tx-4 (cal29a) with a subtle change in position 20 and 21 inverting the amino acid DK for KD; this suggests that subtle changes in the conotoxin compositions (with the possible help of specific enzymes) allow the snail to use the same gene for different conopeptides that have different functions, without involving differential splicing.

A framework of eight cysteines (CCC-C-CC-C-C) assigned as 29, was described as an O1 superfamily member. Additionally, the presence of two new superfamilies in the venom of *C. californicus* was reported, broaden the presence of the O-superfamily in this species.

An extensive study of the complete transcriptome of this specie could allow us to identify if more superfamilies are different to the ones that have been described and are present in the venom of *C. californicus*.

Furthermore, this work reinforces the idea of recognizing the strong potential of O1 superfamily members present in *C. californicus* on the biomedical field. However, we still have a long way to go in the understanding and biological characterization of the O2 and O3 superfamilies.

## 5. Materials and Methods

### 5.1. Venom Purification

Specimens of *C. californicus* were collected from a sandy sublittoral area of Ensenada, Baja California, Mexico. The venom ducts were dissected and immediately homogenized on ice, in 1 mL of 40% (*v*/*v*) acetonitrile (ACN) (Fermont, Monterrey, NL, México) containing 0.1% (*v*/*v*) trifluoroacetic acid (TFA) (Fluka, St. Louis, MO, USA). The homogenate was centrifuged at 10,000 × *g* for 10 min, at 4 °C and the supernatant was lyophilized and stored at −80 °C until peptide purification.

### 5.2. Peptide Purification 

The solubilized venom was fractionated by means of RP-HPLC (Agilent 1220 Series LC System), with an analytical C18 Zorbax 300SB column (4.6 × 250 mm, 5 μm particle size) and a Zorbax 300SB C18 pre-column (4.6 × 12.5 mm, 5 μm particle size) previously equilibrated in a solution of 0.12% (*v*/*v*) TFA (Solution A). The total venom retained in the pre-column was desalted with the same equilibration solution at a flow of 1.0 mL/min over 5 min. Fractions of the venom components were collected every 5 min and were eluted with a linear gradient from 0% to 60% (*v*/*v*) of pure ACN containing 0.10% (*v*/*v*) of TFA (Solution B), over 65 min, at a flow rate of 1.0 mL/min. Each fraction was tested in a cell growth inhibitory assay with reference strain H37Rv. The chromatographic fraction containing the peptide of interest was re-purified, via RP-HPLC, using a linear gradient, from 13% to 37% (*v*/*v*) of Solution B, over 60 min, at a flow rate of 1.0 mL/min. All purification steps were conducted at room temperature and the absorbance was monitored at 230 nm. The purified peptide was lyophilized, named O1_cal29b according conventional nomenclature [23], and evaluated in two MDR strains. Deionized water was purified using a Milli-Q system (Pure Lab Flex, Elga from Ion Torrent, Life Technologies, Grand Island, NY, USA).

### 5.3. 3´RACE

Ten venom ducts from *C. californicus* were dissected and immediately submerged in RNAlater solution (Quiagen) before total RNA extraction. This step was performed with TRI reagent (Sigma Aldrich, St. Louis, MO, USA), according to the manual instructions. Approximately 1 µg of total RNA was used to generate cDNA using a RLM-RACE kit (Ambion, from Invitrogen, Carlsbad, CA, USA). For polymerase chain reaction (PCR) amplification of the gene encoding O1_cal29b, degenerated oligonucleotide primers were designed based on the partial mature peptide sequence; 5´ outer primer (5´-AGG CCT AAA TGY TGT TGT GTS-3´) 5´ inner primer (5´-TGT GGC GTS GTS GGC AGG AAA-3´); where Y = C or T, S = G or C. The reverse primers were provided with the kit. The PCR amplification was carried out using cycling kit protocol and the PCR products were analyzed by the gel electrophoresis. The amplified fragments were purified and ligated into the T-tailed plasmid pGEM-T vector (Promega). Subsequently, the ligation products were transformed into competent cells of *Escherichia coli* DH5α. Transformed colonies were screened by white-blue identification for sequence analysis. Plasmids containing expected inserts were sequenced and sequences were analyzed using the Basic Local Alignment Search Tool (BLAST) [24], where the 232 bp DNA fragment was identified as our product of interest.

### 5.4. 5´RACE

Based on the 3’ partial sequence determined by 3’ RACE, the anti-sense specific primers for 5’ RACE were designed and synthesized as follows: gene-specific outer primer (5´-GAA GGA CGG ATA GGA AGA AGG-3´) corresponding to its untranslated region C-terminal, and gene-specific inner primer (5´-GCC ACT GCT GGG GCA AGG AAG-3´) corresponding to amino acid sequence (LPCPSSG). The amplified products of 5´-end cDNA of O1_cal29b were cloned into the pGEM-T vector for sequencing. The signal peptide sequence of the conotoxin precursor was predicted online with SignalP 4.1 Server [10].

### 5.5. Transcriptome of the Venom Gland

#### 5.5.1. Total RNA Extraction

Fifteen specimens of *C. californicus* were collected in Ensenada, Baja California, on the Pacific coast of Mexico. The venom ducts were dissected from each cone snail under RNAse-free conditions and pulled together in a single tube. The RNA was isolated using the SV Total RNA Isolation System of Promega following the protocol of the manufacturer. 

#### 5.5.2. RNA-Seq Library and Venom Duct Transcriptome Assembly

A complementary DNA (cDNA) library was constructed using the Illumina TruSeq Stranded mRNA Sample Preparation Kit (Illumina, San Diego, CA, USA). DNA sequencing was performed at the Core Facility of the Institute of Biotechnology in Cuernavaca, Mexico, with a Genome Analyzer IIx of Illumina, using a 72 bp paired-end sequencing scheme over cDNA fragments ranging in size of 200–400 bp. The library consisted of two fastq files, from which the adaptors were clipped-off. The quality of cleaned raw reads was assessed using the FastQC program. Since no reference genome is available for the examined Conus snail, short reads were assembled into contigs in a de novo fashion with Trinity software [25] (v. 2.0.3), using the standard protocol [26], executing the strand-specific parameter and normalizing reads. To weigh the quality of the assembly, basic statistics for the number of genes and isoforms as well as the contiguity were obtained by running the TrinityStats.pl script. 

#### 5.5.3. Bioinformatic Analysis of Conotoxin Identification

After the sequence assembly, a database of open reading frames longer than 50 amino acids was generated using the Transdecoder utility included in Trinity. Moreover, the translated peptide sequences were filtered by removing redundant sequences and keeping sequences that begin with methionine and a signal peptide. The signal peptide sequences were determined by using the signal P4.1 [27] 

In order to discover new conotoxins from the filtered sequences, a profile of the hidden Markov model (pHMM) was constructed for each conotoxin superfamily. The pHMMs were built following a similar methodology used in Robinson (2014) [28] and Peng (2016) [29]. According to this methodology, the sequences of each conopeptide superfamily are aligned, for this purpose, we used MAFFT 7.0 [30]. Then the multiple alignments of the sequences were used as input to generate the pHMMs by using HMMER 3.0 [31]. The profiles allowed us to identify the conotoxins presented in the transcriptome. For practical purposes, only the results of the O-superfamily are shown.

### 5.6. Antimycobacterial Susceptibility Assay

Antimycobacterial testing was performed using a colorimetric CellTiter 96^®^ Aqueous One Solution Cell Proliferation Assay (Promega, Madison, WI, USA). The test was performed in 96-wells sterile microplates. All wells were added 100 μL of Middlebrook 7H9 Broth (Becton Dickinson, Franklin Lakes, NJ, USA), supplemented with 0.2% (*v*/*v*) glycerol (Sigma-Aldrich, St. Louis, MO, USA), and 10% (*v*/*v*) oleic acid, albumin, dextrose and catalase (OADC, Edmond OK, USA; Becton Dickinson, Franklin Lakes, NJ, USA). The tested strains were H37Rv (ATCC 27294), reference-susceptible strain, and two MDR strains. The MDR-1 is a strain resistant to Streptomycin and Isoniazid, and MDR-2 is a strain resistant to three first-line drugs Isoniazid, Rifampicin, and Pyrazinamide. The inoculum was prepared from fresh Lowenstein Jensen medium and was resuspended in Middlebrook 7H9 Broth. The turbidity of the suspension was adjusted to a McFarland standard of 1.0. The suspension was homogenized and allowed to precipitate larger particles. The supernatant was diluted at a ratio of 1:20, and 100 μL was used as inoculum. To reduce evaporation from the plates, 200 μL of sterile water was added to all outer perimeter wells. Each microplate was incubated for 7 days at 37 °C. Following incubation, 10 μL of CellTiter 96^®^ solution (158 μL/mL) was added to each well. The plates were re-incubated at 37 °C for two to four hours. The absorbance, recorded at 450 nm using an Epoch Microplate Spectrophotometer, was used to indicate bacterial growth. In order to know the maximal growth reference, growth controls (GC) with 100 μL of inoculum plus 100 μL of medium were used. First line drugs against tuberculosis, such as Isoniazid (INH), Rifampicin (RIF), Ethambutol (EMB), Streptomycin (SM) and Pyrazinamide (PZA) was used as positive controls.

In the first screening, several fractions of *C. californicus* venom were evaluated at 200 μg/mL each with H37Rv strain. Sub-fractions were screened at 100 to 12.5 μg/mL (data not shown) and the peak that contains the conotoxin of our interest was tested at 12.5 μg/mL (3.52 μM) in reference strain. Minimal inhibitory concentration (MIC) was determined with multi-drug resistant strains, as described below.

### 5.7. Minimal Inhibitory Concentration (MIC) Assay

The MIC of conotoxin was tested in 96-well sterile microplates. All wells received 100 μL of supplemented Middlebrook 7H9 broth. One-hundred microliters of a 4× working solution of O1_cal29b were added to the first row of each column. One-hundred microliters were transferred from row 1 to row 2, and the contents of the wells were homogenized by pipetting. Identical serial 1:2 dilutions were continued through the rows, and 100 μL of the excess medium was discarded from the well in the final row. Subsequently, 100 μL of *M. tuberculosis* inoculum was added to the wells. The final test concentration range was 3.52–0.22 μM. Each experiment was performed in triplicate.

## Figures and Tables

**Figure 1 toxins-11-00128-f001:**
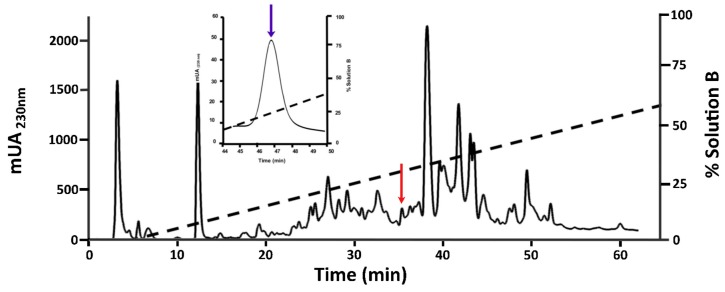
Purification and identification of O1_cal29b conotoxin from *C. californicus* using a linear gradient (0% to 60% of Solution B). The red arrow indicates the elution time for the conotoxin O1_cal29b (specifically between minutes 35 to 37). This fraction has activity against *M. tuberculosis* H37Rv. The fraction with the red arrow was further re-purified (inset) with a slower linear gradient (13 to 37% of Solution B, the segment is shown on top, blue arrow). In both chromatographic profiles, the broken lines indicate the linear gradient of solution B. The resulting peak (blue arrow) was evaluated in biological assays against two multidrug resistent Mtb strains and H37Rv.

**Figure 2 toxins-11-00128-f002:**
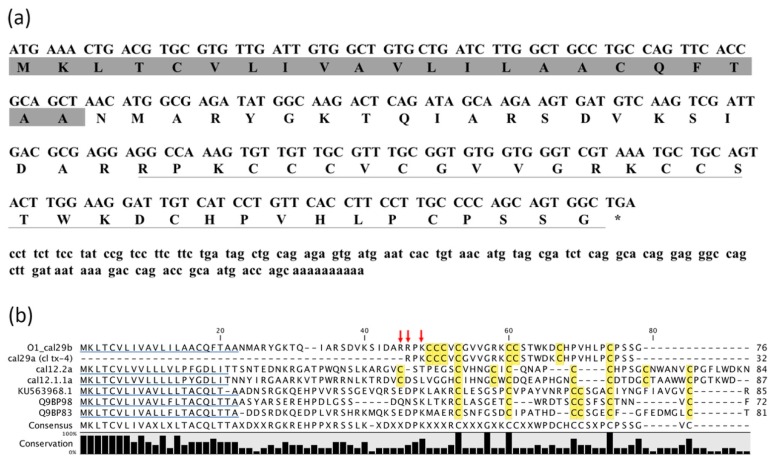
Gene sequence of *O1_cal29b*. (**a**) The cDNA sequence and cDNA-encoded precursor of conotoxin. The coding region of cDNA is shown in capital letter. The signal peptide sequence is grey shadowed and the mature peptide sequence is underlined. Between the signal sequence and the mature toxin region is the propeptide region. *represents the stop codon. (**b**) Alignment for O1_cal29b precursor with similar conopeptides belonging to the O1-superfamily. The GenBank accession numbers were used for conotoxin alignment: KU563968.1 from *Conus betulinus*; Q9BP83 from *Conus arenatus*; Q9BP98 from *Conus ventricosus*; cl tx-4 (cal29a) [1] and cal12.2, cal12.1.1a from *C. californicus* [2]. Signal peptides are underlined in blue, Cys residues in mature peptides are highlighted in yellow, and the conservation percentage for each amino acid is shown in bar format at bottom of the figure. The red arrows indicate three possible sites for N-terminal processing, according to the mechanism proposed by Dutertre et al (2013) [4].

**Figure 3 toxins-11-00128-f003:**
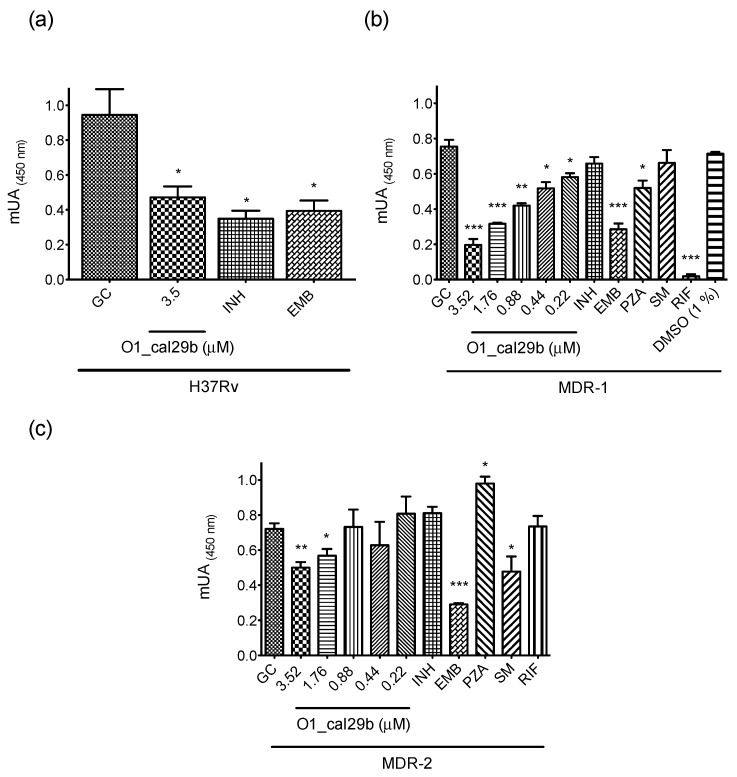
(**a**) Growth-inhibitory effect of O1_cal29b peptide against pathogenic *M. tuberculosis* (H37Rv strain); (**b**) minimal inhibitory concentration (MIC) cut-off values for multi-drug resistant strain MDR-1, a strain resistant to Streptomycin and Isoniazid; (**c**) MIC evaluation for MDR-2, strain resistant to Isoniazid, Pyrazinamide, and Rifampicin. The MIC range tested was between 3.52–0.22 μM of conotoxin in both MDR strains. First-line drugs concentration tested were: INH 0.9 μM, EMB 4.9 μM, RIF 0.6 μM, SM 1.7 μM, PZA 101.5 μM. The statistical significance of differences between treatments and growth control were analyzed using a Student´s test. * *p* < 0.01, ** *p* < 0.001, *** *p* < 0.0001 vs Growth Control (GC).

**Table 1 toxins-11-00128-t001:** O-superfamily sequences from *C. californicus* venom duct transcriptome. Sequences have been clustered by the O1, O2 and O3-superfamilies, according to their signal peptide. Cys residues in mature peptides are bolded and underlined. **a**) O1 superfamily sequences already described before; **b**) novel O1, O2 and O3 superfamily sequences described in this work. Colored in red are the possible cleavage sites for the mature toxins predicted by ProP 1.0 Server. Highlighted in yellow, the amino acids change in the toxin cal12.2e compared with reported toxin cal12.2c [2].

**(a)**				
**Name**	**Cysteine Framework**	**Signal peptide**	**Toxin**	**GenBank #**
O1_cal1.2	I	MKLTCVFIIAVLILTACHFIVAD	AG**CC**PTIMYKTGA**C**RTNR**C**	ADD97803.1
O1_cl6b/cl6.2	VI/VII	MKLTCVLIIAVLILTACQFIAAD	N**C**IPKNHF**C**GLLHHSRN**CC**TPT**C**LIV**C**F	ADB93121.1
O1_cal6.1b	VI/VII	MKLTTVLVVALLVLAACQFTVTD	**C**LAGSAR**C**EFHKPST**CC**SGH**C**IFWW**C**A	ADB93119.1
O1_cl6.6a	VI/VII	MKLTCVLIAAVLLLAVCQLDSADAT	TRG**C**KSKGSF**C**WNGIE**CC**GGN**C**FFA**C**IY	ADB93125.1
O1_cl6.10	VI/VII	MKLTCVLIAAVLLLAVCQLDSADAT	TRG**C**KTKGTW**C**WASRE**CC**LKDCLFV**C**VY	ADB93112.1
O1_cl6.5	VI/VII	MKLTCVLIVAVLVLTACQFTAAI	**C**IPDHHG**C**GLLHHSRY**CC**NGT**C**FFV**C**IP	ADB93124.1
O1_cal6.1a	VI/VII	MKLTTVLVVALLVLAACQFTVTD	**C**LAGSAR**C**EFHKPSS**CC**SGH**C**IFWW**C**A	ADB93120.1
O1_cl6.3	VI/VII	MKLTTVLIVAVLVLAACQFTVTD	GLSRPSKG**C**IGGGDP**C**EFHRGYT**CC**SEH**C**IIWV**C**A	ADB93122.1
O1_cal6.1e	VI/VII	MKLTTVLIVAVLVLAACQFTVTD	**C**IGGGDP**C**EFHRGYT**CC**SEH**C**IIWV**C**A	ADB04242.1
O1_cal6.1a	VI/VII	MKLTTVLVVALLVLAACQFTVTD	**C**LAGSAR**C**EFHKPSS**CC**SGH**C**IFWW**C**A	ADB93120.1
O1_cl6.6b	VI/VII	MKLTCVLIAAVLLLAVCQLDSADAT	TRG**C**KSKGSF**C**WNGIE**CC**GGN**C**FFA**C**VY	ADB93126.1
O1_cal12.1.3a	XII	MKLTCVLVVLLLLLPYGDLI	DV**C**DSLVDGR**C**IHNG**C**F**C**EESKPNGN**CC**DTGG**C**VWWW**C**PGTKWD	ABR92964.1
O1_cal12.2a	XII	MKLTCVLVVLLLLLPYGDLI	GV**C**STPEGS**C**VHNG**C**I**C**QNAP**CC**HPSG**C**NWANV**C**PGFLWDKN	ABR92966.1
O1_cal12.1.2b	XII	MKLTCVLVVLLLLLPYGDLI	DV**C**DSLVDGR**C**IHNG**C**Y**C**ERDAPNGN**CC**NTDG**C**TARWW**C**PGTKWD	ABR92953.1
O1_cl12.3	XII	MKLTCVLVVLLLFLPYGDLI	DV**C**DSLVGGN**C**IHNG**C**W**C**DQEAPHGN**CC**DTDG**C**TAAWW**C**PGTKWD	ADB93095.1
O1_cal12.1p1	XII	MKLTCVLVVLLLLLPYGDLI	DV**C**KKSPGK**C**IHNG**C**F**C**EQDKPQGN**CC**DSGG**C**TVKWW**C**PGTKGD	AEC22829.1
O1_cal12a	XII	MKLTCVLVVLLLLLPYGDLI	DV**C**DSLVGGH**C**IHNG**C**W**C**DQEAPHGN**CC**DTDG**C**TAAWW**C**PGTK	P0DJC1.1*
**(b)**				
**Name**	**Cysteine Framework**	**Signal peptide**	**Toxin**	
O1_cal6.18	VI/VII	MKLTYVLIVAMLVLVVCRAD	**C**FGRGGL**C**TWFDPSV**CC**SGI**C**TFVD**C**W	
O1_cal6.19	VI/VII	MKVTCVLVLTLMALTVCQVATAY	**C**INVGM**C**IYDGY**CC**SNR**C**WGGM**C**SPWR	
O1_cal6.20	VI/VII	MKLTCVLIVAVLILTACQVIAAD	GWFGEESS**C**WW**C**TGQNK**CC**EEAQV**C**QSVNYA**C**PPARR	
O1_cal6.21	VI/VII	MQLTHVLVVGLLVLTSFQPINAV	TNRVD**C**SAPEDKSEPGYW**C**GLEPL**CC**YSGK**C**FVI**C**FGSKPAGT	
O1_cal6.22	VI/VII	MKLTCVLIVAVLILTACQVIAAD	EADANRLSTRW**C**A**C**GVNYY**CC**NEV**C**TWREDP**C**P	
O1_cal6.23	VI/VII	MKLTAVLMVAVLVLTACQLITAN	E**C**SRKGEW**C**GLESVL**CC**NGGSWN**C**WFV**C**TA	
O1_cal6.24	VI/VII	MKLTCVMIVAVLVLTVCKVVTSD	QLKKLRRE**C**YLEPGDS**C**FHDDGRGA**CC**EGT**C**FFGVA**C**VPWS	
O1_cal6.25	VI/VII	MKLTHVLIVAVLVLTVCHLTMAV	**C**KSGGQA**C**WFLLKKHN**CC**SGY**C**IVAV**C**AG	
O1_cal6.26	VI/VII	MKLTCVMIVAVLVLTVCKVVTSD	QLKKLRRE**C**YLEPGDS**C**FHDDGRGA**CC**EGT**C**LFGIN**C**VASW	
O1_cal6.27	VI/VII	MKLTCVLIAAMLLLAVCQLDSADAT	ETG**C**KKDGSW**C**WIPSE**CC**IES**C**LIT**C**WY	
O1_cal6.28	VI/VII	MKLTCVLIVAVLILTACQVIAAD	EATNRATKRG**C**LM**C**WGSNVR**CC**EKANA**C**VSINYE**C**PKARR	
O1_cal6.29	VI/VII	MKLTCVLIVAVLVLTACQFTAAI	SQTQRLSKK**C**IEDNHA**C**GLLHHSPY**CC**NGT**C**FIV**C**IP	
O1_cal6.30	VI/VII	MKVTCVLTLAVLILTIGQIANAD	STLGQRY**C**KASGSW**C**GIHKHRE**CC**SGN**C**FFW**C**VYNGK	
O1_cal6.31	VI/VII	MKLTCVLIAAVLLLAVCQLDSADAI	TRD**C**KTKGYA**C**FASTE**CC**VQD**C**WLV**C**LY	
O1_cal6.32	VI/VII	MKLTCVLIVSVLILTACQFTAAV	D**C**HSTGYL**C**FWWHE**CC**SNF**C**IPLQQR**C**F	
O1_cal6.33	VI/VII	MKLTCVVIIAVLILTACQFTTAD	D**C**KPKNNL**C**LWSSE**CC**SGI**C**FPFAQR**C**T	
O1_cal6.34	VI/VII	MKLTCVLIVAVLILTACQVIAAD	SS**C**WF**C**STGFNK**CC**ESTGD**C**MTYPSEYNAS**C**PEA	
O1_cal6.35	VI/VII	MKLTCVLIVAVLILTACQVIAAD	EAEATNRAIKRGWFGEESS**C**WW**C**TGFNK**CC**EAAAV**C**QSVNSA**C**P	
O1_cal6.36	VI/VII	MKVTCVLTLAVLILTVGQMVTAD	**C**RSPGSW**C**FYKHSN**CC**SGN**C**FLW**C**VQNGK	
O1_cal6.37	VI/VII	MKLTCVMIVAVLLLTVCKVVTSD	QLKKLRRE**C**YLEPGDS**C**FHHDGRGA**CC**EGT**C**FFGVA**C**VPW	
O1_cal6.38	VI/VII	MKLTFVLIVAVLVLAVCNFTVAD	KANNAEAPEQEKRA**C**TPNGSY**C**NILSGKLN**CC**SGW**C**LALI**C**AG	
O1_cal12b	XII	MKLTCMLVVLLLVLPFGDLI	ANTGGL**C**GMPPGV**C**YPNG**C**A**C**GQDTP**CC**HPSG**C**NRYNY**C**GPLLE	
O1_cal12c	XII	MKVTCVLVVLLLLLPYGDLLGN	SV**C**DFGS**C**VHNG**C**Y**C**EEHRP**CC**TPGS**C**SSWWPR**C**PGSMMDP	
O1_cal12.2e	XII	MKLTCVLVVLLLVLPFGDLI	GV**C**STPEGS**C**VHNG**C**I**C**QNAP**CC**HPSG**C**NWVNV**C**PGFLWDRS	
O1_cal29b	XXIX	MKLTCVLIVAVLILAACQFTAAN	RPK**CCC**V**C**GVVGRK**CC**STWKD**C**HPVHLP**C**PSSG	
O2_cal30	XXX	MEKLIILLLVASLLVTTDSVVKGK	KAARGWLFNEVET**C**ELGGLGDP**C**SGSGD**CCC**DQ**C**L**C**SGSYEH**C**TQNPDRWF**CC**RTYGN	
O3_cal14d	XIV	MFRLGVFLLTFLLLVSMATSE	YSRGRIMARASE**C**VNE**C**VESGHNTFH**C**ERH**C**SNT	
O3_cal6.1a	VI/VII	MSGSGAMLLGLLILVAMAT	SLDTREI**C**WNHSE**C**DDPSEW**CC**RMGSGHGS**C**LPV**C**RP	
O3_cal6.1b	VI/VII	MSGSGAMLLGLLILVAMAT	SLDTREI**C**WNHSE**C**DDPSEW**CC**RMGSGHGS**C**QPV**C**RP	
O3_cal6.1c	VI/VII	MSGSGAMLLGLLILVAMAT	SLDTREI**C**WHQSE**C**DDPNEW**CC**IMGTSYGS**C**QPV**C**RP	
O3_cal6.2	VI/VII	MSGSGVLLLTLLLLVPLSAL	AKE**C**SMYY**C**SGGDF**CC**PGLK**C**GDPTGKKI**C**IEPGK	
O3_cal6.3	VI/VII	MSGTTVLLLTCLFLVTMAT	SD**C**DLYDDS**C**TGTEI**CC**TPPGDYQGN**C**MEGED**C**PSGGR	
O3_cl6d	VI/VII	MSGTGVLLLTLLLLVTMATSD	DA**C**SLLNGDD**C**GPGEL**CC**TPSGDHQGT**C**ETS**C**W	
O3_cal27	XXVII	MSGTGVLLLTLLLLVAMAASD	MLSSLIQAHERDSEES**C**KSYGGGP**C**PSGED**CCC**PPGRSTGT**C**KRT**C**NNGSV**C**A	
O3_contryphan-like cal1		MTRTAVLLLTLLFLVAMAASD	KIKTREV**C**WTEEE**C**ENWE	
O3_contryphan-like cal2		MTRTAVLLLTLLFLVAMAASD	KIKTREL**C**WTEEE**C**ENWE	
O3_contryphan-like cal3		MTRTAVLLLTLLFLVAMAASD	KIKTREL**C**WSERE**C**ENGK	
O3_contryphan-like cal4		MTRTAVLLLTLLFLVAMAASD	KIKTREV**C**WNEEE**C**ENWE	

* UniProtKB/Swiss-Prot number.

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
