# Peer review of "The Diversified O-Superfamily in Californiconus californicus Presents a Conotoxin with Antimycobacterial Activity"

_toxins, 2019, doi:10.3390/toxins11020128_

Round 1

Reviewer 1 Report

The article presents a new conotoxin with an antimycobacterial activity. Morover, through transcriptomic studies authors were able to isolate an additional group of conotoxins. They also propose nomenclature for newly found conotoxins and classify to a Cys-framework. Its bioactivity is at the same range with drugs available on market and applied for Mtb treatment.

This article is comprehensive to read and demonstrates new candidates for development of peptide-based drug leads for Mtb treatment.

Taking in account all mentioned above I would like to recommend this article for publication in Toxins as it is.

Author Response

We appreciate comments of reviewer one, we are grateful and honored with the decision.

Reviewer 2 Report

In this manuscript, the authors have analyzed the Californiconus californicus venom gland transcriptome and report several putative members of O2 and O3 superfamily conotoxins. This study has also identified a new conotoxin assigned to O1-superfamily capable of inhibiting the growth of M. tuberculosis. This is a well-written manuscript and from my point of view, this manuscript reaches the publication level of this journal.

Author Response

We appreciate comments of reviewer two, we are grateful and honored with the decision.

Reviewer 3 Report

Line 65–66: Where is the result of the antimycobacterial assay?

Line 67–68, Figure 1: I am a bit confused here. The resulting peak fraction (used for biological assays) should be the peak as shown in the inset, not the peak indicated by the red arrow as in the figure. Please move the red arrow to the inset.

Figure 2, line 100-106: I would suggest adding the sequence of cl tx-4 (cal29a) in Figure 2b for clarification.

Figure 3a-c: Change (mM) to (uM).

Line 238: change "specie" to "species". 

Author Response

Comments of reviewer three.

Line 65–66: Where is the result of the antimycobacterial assay?

Answer: we had these results as supplementary figure 1 and is now mentioned in text

Line 67–68, Figure 1: I am a bit confused here. The resulting peak fraction (used for biological assays) should be the peak as shown in the inset, not the peak indicated by the red arrow as in the figure. Please move the red arrow to the inset.

Answer: Done.

Figure 2, line 100-106: I would suggest adding the sequence of cl tx-4 (cal29a) in Figure 2b for clarification.

Answer: Done

Figure 3a-c: Change (mM) to (uM).

Answer: we do have uM, however, we think that probably reviewer three use a different computer system that the one we use, and the (m in symbol form) change to a regular m. we are going to be careful with this issue in the final document. Thanks for the observation.

Please find a screenshot of the original figure legend:

Line 238: change "specie" to "species". 

Answer: Done